# Changing Susceptibility of Staphylococci in Patients with Implant-Based Breast Reconstructions: A Single-Center Experience

**DOI:** 10.3390/medicina58081130

**Published:** 2022-08-20

**Authors:** Hyo Young Kim, Hyung-Suk Yi, Jeong-Jin Park, Seok-Kyung In, Hong-Il Kim, Jin-Hyung Park, Woon-Hyoung Lee, Yoon-Soo Kim

**Affiliations:** 1Department of Plastic and Reconstructive Surgery, Kosin University, College of Medicine, 262 Gam-cheon-ro, Seo-gu, Busan 49267, Korea; 2Department of Plastic and Reconstructive Surgery, Bundang Seoul University, College of Medicine, 173 Gumi-ro, Bundang-gu, Seongnam 13620, Korea; 3Department of Laboratory Medicine, College of Medicine, Kosin University College of Medicine, 262 Gam-cheon-ro, Seo-gu, Busan 49267, Korea

**Keywords:** mammaplasty, capsular contracture, bacteria, antibacterial agents

## Abstract

*Background and Objectives*: Infections and capsular contractures remain unresolved issues in implant-based breast reconstruction. Capsular contractures are thought to be caused by the endogenous flora of the nipple duct. However, little is known about the antibiotic susceptibility of the microorganisms involved. This study aimed to evaluate the composition of endogenous breast flora and its antimicrobial susceptibility in patients with breast cancer. This study will aid in selecting a prophylactic antibiotic regimen for breast reconstruction surgery. *Materials and Methods*: We obtained bacteriologic swabs from the nipple intraoperatively in patients who underwent implant-based breast reconstruction following nipple-sparing mastectomy between January 2019 and August 2021. Antibiotic susceptibility tests were performed according to the isolated bacteriology. Statistical analysis was performed based on several patient variables to identify which factors influence the antibiotic resistance rate of endogenous flora. *Results*: A total of 125 of 220 patients had positive results, of which 106 had positive culture results for coagulase-negative Staphylococcus species (CoNS). Among these 106 patients, 50 (47%) were found to have methicillin-resistant staphylococci, and 56 (53%) were found to have methicillin-susceptible staphylococci. The methicillin resistance rate in the neoadjuvant chemotherapy group (56.3%) was significantly higher (OR, 2.3; *p* = 0.039) than that in the non-neoadjuvant chemotherapy group (35.5%). *Conclusions*: Based on the results, demonstrating high and rising incidence of methicillin-resistant staphylococci of nipple endogenous flora in patients with breast cancer compared to the past, it is necessary to consider the selection of prophylactic antibiotics to reduce infections and capsular contracture after implant-based breast reconstruction.

## 1. Introduction

The rate of breast reconstruction following mastectomy is on the rise. Using either autologous tissue or an implant carries a risk of infection [1]. Moreover, infection following breast reconstruction remains an unresolved issue. Infections can lead to complications, ranging from mild (superficial cellulitis) to severe (repeated procedures for implant failure) to life-threatening sepsis [2]. Subclinical infection has been linked with an increased risk of capsular contracture, which can cause significant deformity, pain, and distress in the patient [3,4,5].

The average incidence of surgical site infections following breast surgery, especially in implant-based breast reconstructions, has been reported to be 0.4–17% worldwide [6]. Current guidelines for the prevention of surgical site infection recommend administering a first-generation cephalosporin once before surgery as prophylaxis [7,8]. Despite the prophylactic use of first-generation cephalosporins, infections continue to occur, and the probability of capsular contracture increases with time (2.8–20.4%) [9,10,11,12,13]. Immediate infection after breast reconstruction not only entails implant replacement, an increase in hospitalization period, and economic problems, but also delays chemotherapy, thereby affecting oncologic treatment [14]. During the delayed period, capsular contracture causes pain, stiffness, and cosmetic deformity, necessitating revision surgery. Capsular contracture is associated with subclinical infection, and it is known that subclinical infection is caused by contamination of the implant surface with endogenous flora [5,15,16,17]. Endogenous flora leads to infection when skin integrity is compromised [18]. The predominant bacterial species found in the breast are Staphylococcus epidermidis and Propionibacteriaceae [8,19]. *S. epidermidis* is a significant pathogen in implant-based breast reconstruction because it is associated with device-related infections [20]. In addition to the study carried out in 1988 on the antibiotic susceptibility of Staphylococcus identified on the mammary gland skin surface of pregnant women, there are few recent data on the antibiotic susceptibility of Staphylococcus in breast cancer patients who underwent implant-based breast reconstruction as well as insufficient information on the resistance rate to various antibiotics [21].

Therefore, the authors sought to investigate the breast endogenous flora of patients who underwent implant-based breast reconstruction at a single institution and confirm their antibiotic susceptibility. In addition, we examined several factors that increase the prevalence of antibiotic resistance in the endogenous flora. Our research will aid in the selection of a prophylactic antibiotic regimen during surgery and empirical antibiotic regimen in the event of an infection.

## 2. Materials and Methods

### 2.1. Study Design

Between January 2019 and August 2021, 220 female patients from Busan, Korea, who had undergone direct-to-implant breast reconstruction following nipple-sparing mastectomy and provided informed consent were included in the study. Nipple swabs were collected and sent to the Department of Microbiology for microbiological and antibiotic susceptibility testing. The mean patient age at the time of surgery was 49.3 years (range: 27–68 years). To identify the factors that can influence antibiotic resistance of endogenous breast flora, we recorded patient age, body mass index (BMI), and history of neoadjuvant chemotherapy (Table 1). To identify the factors affecting the antibiotic resistance rate, patients were divided into a methicillin resistance group and a methicillin-susceptible group and age, body mass index (BMI), and history of neoadjuvant chemotherapy were compared. This study was approved by the Institutional Review Board of Kosin University Gospel Hospital (IRB No. 2020-11-014-002) and was performed in accordance with the principles of the Declaration of Helsinki. The patients provided written informed consent for publication and use of anonymized images.

### 2.2. Perioperative Procedures

Preoperative antibiotic prophylaxis with 2 g of a first-generation cephalosporin (cefazedone, Yooyoung Pharmaceutical Co., Jincheon, South Korea) was administered intravenously at least 30 min before surgery. For each patient, disinfection of the chest wall and nipple/areola area with povidone-iodine solution was performed. Gentle swabbing was performed on the nipple/areolar area using a dry cotton swab before mastectomy (Figure 1). We collected 220 swabs and sent them for bacterial culture, after which reconstructive surgery was performed.

### 2.3. Microbiology and Microbial Susceptibility

The swab was stored in an Amies transport medium as a routine procedure and sent to the laboratory for Gram staining and culture. To shake out the bacteria, the swab head was immersed in a sterile saline solution and centrifuged to collect the bacterial pellet. Bacterial species were identified by matrix-assisted laser desorption/ionization time-of-flight mass spectrometry (MALDI-TOF MS; Bruker Daltonics, Bremen, Germany). The antimicrobial susceptibilities of the clinical isolates were determined using VITEK AST cards (bioMérieux).

### 2.4. Statistical Analysis

To identify the factors affecting the antibiotic resistance rate, patients were divided into two groups: a methicillin resistance group (*n* = 51) and a methicillin-susceptible group (*n* = 74). Student’s *t*-test was used to analyze age and BMI in both groups. The chi-square test was performed to assess the antibiotic resistance rate variables as factors for neoadjuvant chemotherapy. The data were analyzed using Microsoft Excel version 16.53 (Microsoft Corp., Redmond, WA, USA). Statistical analysis was conducted using SPSS version 18.0 (SPSS Inc., Chicago, IL, USA). In all statistical comparisons, a *p*-value of <0.05 was considered to indicate statistical significance.

## 3. Results

Based on the culture results of the post-disinfection swabs, 125 of the 220 patients (56.8%) yielded positive results for the nipple. A total of 106 patients had coagulase-negative Staphylococcus species (CoNS) (86 patients with *S. epidermidis* (68.8%), 6 with *S. hominis* (4.8%), 5 with *S. lugdunensis* (4%), 3 with *S. warneri* (2.4%), and 5 with other S. spp. (4%)), 1 patient had *S. aureus*, 6 patients had Propionibacterium acnes (4.8%), 5 patients had Micrococcus luteus (4%), and 8 patients had other bacterial spp. (6.4%; Figure 2).

Among patients, 50 (47%) out of the 106 with positive culture results for CoNS were found to have methicillin-resistant staphylococci, and 56 patients (53%) were found to have methicillin-susceptible staphylococci. The methicillin resistance rate in the neoadjuvant chemotherapy group (56.3% (18 out of 32 patients)) was significantly higher (OR, 2.3; *p* = 0.039) than that in the non-neoadjuvant chemotherapy group (35.5% (33 out of 93 patients)) (Table 2). The methicillin resistance rate was not significantly associated with the age (*p* = 0.235) or BMI (*p* = 0.056) of the patients (Table 3). Antibiotic susceptibility tests for methicillin-resistant staphylococci among patients with CoNS-positive culture results are presented in Table 4.

## 4. Discussion

*S. epidermidis* was found in 68.8% of positive cultured cases, demonstrating that this bacterium is a part of the normal flora of the nipple/areola complex and superficial lactiferous duct. Almost half of the breast reconstruction patients had methicillin-resistant staphylococci in their endogenous flora (47%). In patients who received neoadjuvant chemotherapy, the methicillin resistance rate increased to 56.3%. After breast surgery, the most prevalent pathogenic bacteria were *S. aureus* and *S. epidermidis*. *S. epidermidis*, a normal skin flora, is less pathogenic than *S. aureus*. However, it is a common source of infection in line-associated bacteremia and prosthetic device infections. In addition, *S. epidermidis* is the most commonly cultured strain in the biofilms of patients with capsular contracture. Thus, it is important to avoid contamination during surgery [16,22].

As the Guideline for Prevention of Surgical Site Infection recommends a single dose of prophylactic antibiotics prior to surgery, there is an ongoing controversy on the use of prophylactic antibiotics in breast surgery. According to the guidelines, only the first 90 days following surgery are considered infectious [23]. In the context of breast reconstruction, post-radiation infection, delayed infection, and capsular contracture should be included in the category of infection. The infection rate of breast reconstruction is reported to range from 0 to 29% in individual papers with an average of 5.83%. Frequently, additional prophylactic antibiotics beyond those recommended in the guidelines are administered. Looking at other published journals, there is propensity for the guidelines to be disregarded, and there is no standardized protocol [8,24]. It is vital to re-establish guidelines for the precise usage and dosage of prophylactic antibiotics.

Positive results were obtained in 56% of the culture swabs performed after disinfection with povidone-iodine solution before mastectomy. In another breast augmentation study that performed skin swabs in a method similar to that in our study, 38% of nipple swabs were positive [17]. Breast surgery, particularly reconstructive surgery in mastectomy, is a sterile procedure. One study revealed that the bacteria cultured from the skin swab and the capsule of a patient with capsular contracture were identical [16]. In this study, swabs were performed on the superficial layer of the nipple, and it is believed that the bacteria cause direct contamination of the implant surface through the nipple duct.

The prevalence of methicillin-resistant staphylococci is increasing, with the rate reported to be 20% in the community environment and 70% in the hospital environment [25]. In our study, methicillin-resistant staphylococci were found in 56% of the patients receiving neoadjuvant chemotherapy with extensive hospitalization experience compared to 35% of the patients not receiving neoadjuvant chemotherapy with less hospitalization experience. From research in 1987 (0%) to the present (47%), the methicillin resistance rate has been steadily increasing, and this trend is anticipated to continue [25]. It should be recognized that the configuration of the breast endogenous flora has changed and that the operation area can be contaminated by altered endogenous flora even intraoperatively. Therefore, there is a need to redefine the protocol for choosing prophylactic antibiotics. The use of current routine first-generation cephalosporins may not be applicable to hospitalized patients. Therefore, it is necessary to carefully reselect prophylactic antibiotics based on our study for patients with a history of recurrent hospitalization and those with high infection risk [26,27].

Implant-based breast reconstruction should be addressed independently from breast cancer surgery with no reconstruction for prophylactic antibiotics. Unlike other surgical site infections, if infection occurs after implant-based breast reconstruction, devastating complications, such as breast implant replacement and flap necrosis, may ensue. In cases of implant-based reconstruction, the use of acellular dermal matrix (ADM) has increased in recent years. Acellular dermal matrix (ADM) is used for effective breast reconstruction and produces better cosmetic results but may increase the risk of postoperative infection or seroma [14]. Therefore, compared to other surgical procedures, a more sophisticated infection prevention protocol is required. In the case of patients with infection risk factors, patient factors (smoking, age, obesity, hypertension, diabetes mellitus, large breast, immunocompromised state), and surgical factors (ADM use), and neoadjuvant chemotherapy should be added to the prophylactic antibiotic regimen.

Based on the results in Table 4, trimethoprim/sulfamethoxazole is a suitable choice of prophylactic antibiotic because of its resistance rate of 0%; however, its side effects on the kidney and liver should be considered. Vancomycin and teicoplanin are also believed to be effective against breast endogenous flora. Prophylactic antibiotics can be administered to patients with a history of recurrent hospitalization and receiving neoadjuvant chemotherapy. We are currently using prophylactic antibiotics for methicillin-resistant staphylococci among patients with CoNS-positive culture results, including vancomycin and teicoplanin, in the following ways. Vancomycin (1 g) was administered intravenously 30 min preoperatively, and an additional 1 g was administered 12 h later (typically 15–20 mg/kg every 8–12 h for patients) because prophylactic antibiotics are acceptable for up to 24 h [28]. Teicoplanin 400 mg was administered as a single dose prior to surgery. Teicoplanin has a half-life of 45–70 h; therefore, redoses may not be required [29]. In cases of trimethoprim/sulfamethoxazole, there is a lack of studies on the prophylactic efficiency of surgical site injection in trimethoprim/sulfamethoxazole use and no guidelines for the administration of trimethoprim/sulfamethoxazole to patients postoperatively. The use of drugs without criteria had been considered to increase the resistance rate and was not administered [30]. We plan to collect research data on the effects of this regimen by administering prophylactic antibiotics following the abovementioned protocol.

Since our study revealed the resistance rate of endogenous flora, which is the pathogen responsible for capsular contracture, we believe it can be utilized to select prophylactic antibiotics. The limitation of this study is that it was a single-center study based on a single-center experience. Moreover, since culture results were the basis of this study, cases in which the culture result was not identified were excluded; therefore, information on actual Staphylococcus may not be included. In addition, the possibility of some of those Staphylococcus species being external contaminants cannot be excluded. Since this study did not provide data on Staphylococcus, capsular contracture, or postoperative breast infection, it is necessary to carefully interpret these findings. However, endogenous flora has been found to cause capsular contracture and the resistance rate of endogenous flora has increased. Therefore, this study can serve as a guide for the selection of prophylactic antibiotics. Long-term follow-up studies are required to determine the association of different types of prophylactic antibiotics with capsular contracture and infection.

## 5. Conclusions

The methicillin resistance rate of endogenous breast flora, the pathogen responsible for capsular contracture, is increasing. In addition, the rate increased among patients who received neoadjuvant chemotherapy (more hospital exposure). Our study confirmed the recent resistance rate and antibiotic susceptibility of nipple endogenous flora. It is significant in revealing that the methicillin-resistant rate has increased compared to the past. Although there is a limitation of single-center studies in asserting global preoperative antibiotics changes, a change in the selection of preoperative antibiotics can be expected if a larger number of multicenter studies are continued to deepen our understanding of which antibiotics can reduce infection and capsular contracture with fewer side effects.

## Figures and Tables

**Figure 1 medicina-58-01130-f001:**
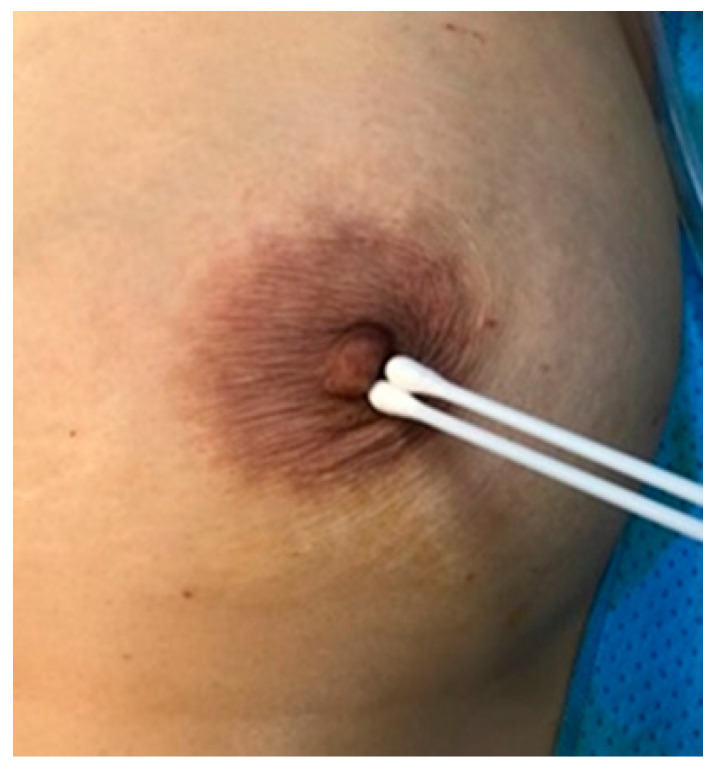
Endogenous flora specimen collection of nipple with cotton swab intraoperatively.

**Figure 2 medicina-58-01130-f002:**
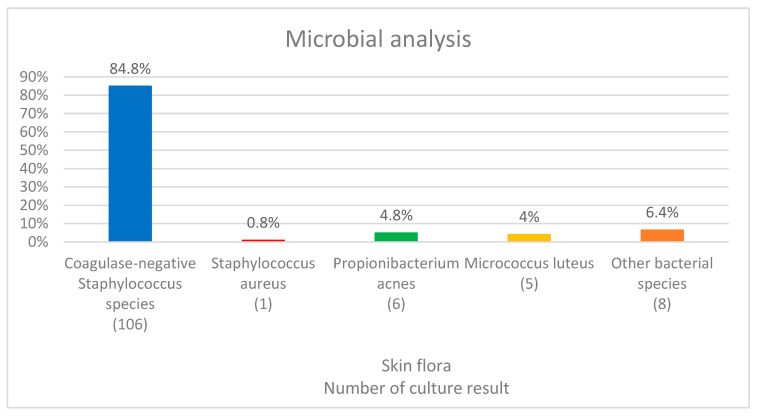
Microbial analysis of nipple/areola complex after disinfection.

**Table 1 medicina-58-01130-t001:** Baseline characteristics of study subjects.

Characteristics	Value (Range, %)
Total number of swabs	220
Age (mean ± SD, y)	49.3 ± 8.3 (27–68)
BMI (mean ± SD, kg/m^2^)	23.6 ± 3.7 (16.5–37.2)
Diabetes (%)	2 (0.9)
Smoking (%)	0 (0)
Neoadjuvant chemotherapy (%)	61 (27.8)

Abbreviations: SD, standard deviation; BMI, body mass index; y, year.

**Table 2 medicina-58-01130-t002:** Statistical analysis for neoadjuvant chemotherapy between the methicillin resistance and methicillin-susceptible groups.

Neoadjuvant Chemotherapy	Methicillin Resistance Group (*n* = 51)	Methicillin-Susceptible Group (*n* = 74)	*p*-Value
Yes	18	14	0.039
No	33	60	

A *p*-value < 0.05 was considered statistically significant.

**Table 3 medicina-58-01130-t003:** Statistical analysis for neoadjuvant chemotherapy between the methicillin resistance and methicillin-susceptible groups.

Characteristic	Methicillin Resistance Group (*n* = 51)	Methicillin-Susceptible Group (*n* = 74)	*p*-Value
Age (mean, y)	46.16	47.71	0.235
BMI (mean, kg/m^2^)	23.9	22.5	0.056

Abbreviation: BMI, body mass index; A *p*-value < 0.05 was considered statistically significant.

**Table 4 medicina-58-01130-t004:** Antibiotic susceptibility test on oxacillin-resistant staphylococci (*n* = 50).

	Susceptible (%)	Resistant (%)
Oxacillin	0	100
Benzylpenicillin	5.9	94.1
Fusidic Acid	14	86
Erythromycin	42	58
Cefoxitin	44.4	55.6
Tigecycline	47.2	52.8
Mupirocin	47.8	52.2
Tetracycline	62	24.5
Gentamicin	66	38
Clindamycin	66	34
Ciprofloxacin	88	12
Teicoplanin	94	6
Telithromycin	96	4
Rifampicin	100	0
Habekacin	100	0
Quinupristin/Dalfopristin	100	0
Linezolid	100	0
Vancomycin	100	0
Nitrofurantoin	100	0
Trimethoprim/Sulfamethoxazole	100	0

## Data Availability

Not applicable.

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
