# Peer review of "Changing Susceptibility of Staphylococci in Patients with Implant-Based Breast Reconstructions: A Single-Center Experience"

_medicina, 2022, doi:10.3390/medicina58081130_

Round 1

Reviewer 1 Report

This study aimed to evaluate the composition of endogenous breast flora and its antimicrobial susceptibility in patients with breast cancer who received implant-based breast reconstruction. It is appropriately conducted in accordance with current guidelines and is correctly designed. It presents different and interesting insights about antibiotic resistance, yet does not propose any practical tips nor suggest "in-house" protocol in cases where antibiotic resistance could be presumed.

Despite this, there are some inaccuracies that need to be revised.

1) at line 122 you refer to Table 1 but the information in the text refers to something different, please provide the necessary correction;

2) at line 18 you refer to Chun YS, Verma K, Rosen H, et al. Implant-based breast reconstruction using acellular dermal matrix and the risk of postoperative complications. Plast Reconstr Surg. 2010;125:429-436 [13] inappropriately. The article is about dual plane immediate implant/expander-based reconstruction ADM assisted vs total or partial submuscular not ADM assisted. The authors do not mention prepectoral reconstruction nor state that the risk of infection is greater in the ADM population. Please, refer to an adequate note.

3) The discussion is well conducted, yet I think it needs some insights: I suggested the authors enrich it with at least their in-house antibiotic prophylaxis and therapy protocol in that population group at the highest infectious risk.

Reviewer 2 Report

Thank you for this manuscript entitled: Changing Susceptibility of Staphylococci in Patients with Implant-based Breast Reconstructions: A Single-Center Experience. I am pleased to review this important study. This study aimed to evaluate the composition of endogenous breast flora and its antimicrobial susceptibility in patients with breast cancer. A total of 125 of 220 patients had positive results, of which 106 had positive culture results for coagulase-negative Staphylococcus species (CoNS). Among these 106 patients, 50 (47%) were found to have methicillin-resistant staphylococci, 25

and 56 (53%) were found to have methicillin susceptible staphylococci. The methicillin resistance rate in the neoadjuvant chemotherapy group (56.3%) was significantly higher (OR, 2.3; p = 0.039) than that in the non-neoadjuvant chemotherapy group (35.5%)

In my opinion your study could be very interesting however the manuscript should be deeply corrected.

In Introduction part:

you should how many surgical site infection are recorded in mammary surgery especially with the implants and what kind of bacterias are responsible for it.

Material and Methods part:

Page 2 line 54

Change „propionibacteria” to Propionibacteriaceae or Cutibacterium.

Line 87 – you write you have collected swabs after mastectomy however the Figure 1 presents as I see in vivo photography - it seems it is before mastectomy, while mamma is not excised yet – please specify, change photo, description etc.

 These part needs improvement. You should in details describe your study design. On page 3, line 99 you wrote “in the two groups” – what two groups? You did not mentioned about two group of patients before in Material and methods part – it should be upgraded. Also please describe – what will you compare in these two groups.

Please describe wider you microbiology methods – were the cotton swabs dry or wet (with what solution?), how did you make bacterial isolation before MALDI identification?

Results part

Figure 2 need to be described more detailed. On Y- axis you put the number of bacterias, as I suppose. In my opinion it would be more representative to show the proporion (%), with accurate number above each bacteria.

Table 1 should be corrected – p-value concerns whole table, it can not be duplicated.

In my opinion table 2 and are not necessary in this study, and if you would like to leave it in the manuscript it should be deeply reedited. Page4, line127 – you mention that in Table 3 MRSA strains are presented – you did not mention that you had 50 MRSA in your study! The figure 2 presents different data.

What is the main problem in these results – there is no information regarding postoperative wound infections – so every further conclusions concerning perioperative antimicrobial prophylaxis can not be done basing your results.

What do you think? Why there are so many positive cultures in your study, while you obtained swabs after disinfection of the chest wall and operative area? Is it your local policy to disinfect nipple-areola area for operation with povidone-iodine solution? Is it based on alcohol solution? Could this particular disinfection method influence your microbiological results? Please discuss these issues in discussion – especially, why there are so many positive swabs after disinfection of the skin (it should be underlined that disinfection does not result the sterility of the skin).

References:

-please check it again - there are some typing errors. 1. 1. ....; 6.6....(...)

-please cite: Palubicka A, Jaworski R, Wekwejt M, Swieczko-Zurek B, Pikula M, Jaskiewicz J, Zielinski J. Surgical Site Infection after Breast Surgery: A Retrospective Analysis of 5-Year Postoperative Data from a Single Center in Poland. Medicina (Kaunas). 2019 Aug 21;55(9):512.

One comment. It is well known that high percent of S.epidermidis is methicillin resitant (MRSE or MRCNS), however they not so often are the cause of postoperative SSI (surgical site infections). This is 180 degrees other than S.aureus! – this is the bug we are afraid. The antimicrobial prophylaxis is designed to prevent S. aureus SSI infections and not S. epidermidis! That is why we use cephalosporins I gen in prophylaxis, because the colonization rate of MRSA is quite low. If the patient is MRSA carrier (this is proved mainly in nasal swabs) this is the reason to change the prophylaxis for example to vancomycin – bu only for this patient not in general. In you will change the general prophylaxis rules basing on you study whis will not change the SSI rate in your patients but in several weeks you will be dealing with high rate of multidrug-resitance in your patients – such things should be avoided in my opinion.

What is interesting – what do you think -  why the neoadivant chth patients are colonized by other resistance patten skin bacterias than the patients who have not chth before surgery: is this related to any factor? Earlier hospital colonization? Earlier antibiotics usage? Or this differences are only statistical error because of small group number etc.?

In my opinion this manuscript should be corrected however after rreevaluation it could be worth to publish it.

Round 2

Reviewer 1 Report

I congratulate the authors because now the manuscript is worth being published. Their contribution is very precious to all surgeons who face breast surgery every day because the problem of antibiotic resistance is becoming more significant and, as we know, it affects the quality of breast implant-based reconstruction. 

Reviewer 2 Report

Hi, thank you for the corrections made in the manuscript. In my opinion it is now much better than the previous version - however I think you should improve the manuscript further. 

The main issues that should be improved:

1. In abstract - the conclusion is not supported by you results at all - in must be changed. There is no support in the manuscript for any conclusion regarding SSI while no data of SSI in you patients group is described.

2. Methods should be rewritten - as I mentioned in the previous review

3. The results part must be improved. I think it is still one major error - you mention MRSA  (without explaining this shortcut)- it is meticillin resitant Staphylococcus aureus. In you results you showed 1 patient with S. aureus (line 119 and Figure 2). However in  line 131 and in Table 4 you mention about 50! S. aureus! Maybe you wanted to write about MSSA? This must be changed.

Additionally tabels and figures must be improved (every shortcut should be explained in legend, you should not put decimal places in number if it is 0 i.e it is 20,00% and should be 20%, 0,00% should be 0% etc.)

4. The conclusions part must be changed - you make conclusions without support of your data. Changes in global perioperative antibiotic prophylaxis should be supported by strong objective data. Please change it. 

Nevertheless, I still thing that this could be a very interesting article and should be published in good Journal, however it must be improved. 
